# Redox Reactions of Biologically Active Molecules upon Cold Atmospheric Pressure Plasma Treatment of Aqueous Solutions

**DOI:** 10.3390/molecules27207051

**Published:** 2022-10-19

**Authors:** Alexander G. Volkov, Jewel S. Hairston, Gamaliel Taengwa, Jade Roberts, Lincoln Liburd, Darayas Patel

**Affiliations:** 1Department of Chemistry and Biochemical Sciences, Oakwood University, Adventist Blvd., Huntsville, AL 35896, USA; 2Department of Mathematics and Computer Science, Oakwood University, Adventist Blvd., Huntsville, AL 35896, USA

**Keywords:** cold plasma, interfacial catalysis, multi-electron reaction, plasma-activated water, reactive oxygen and nitrogen species

## Abstract

Cold atmospheric pressure plasma (CAPP) is widely used in medicine for the treatment of diseases and disinfection of bio-tissues due to its antibacterial, antiviral, and antifungal properties. In agriculture, CAPP accelerates the imbibition and germination of seeds and significantly increases plant productivity. Plasma is also used to fix molecular nitrogen. CAPP can produce reactive oxygen and nitrogen species (RONS). Plasma treatment of bio-tissue can lead to numerous side effects such as lipid peroxidation, genotoxic problems, and DNA damage. The mechanisms of occurring side effects when treating various organisms with cold plasma are unknown since RONS, UV-Vis light, and multicomponent biological tissues are simultaneously involved in a heterogeneous environment. Here, we found that CAPP can induce in vitro oxidation of the most common water-soluble redox compounds in living cells such as NADH, NADPH, and vitamin C at interfaces between air, CAPP, and water. CAPP is not capable of reducing NAD^+^ and 1,4-benzoquinone, despite the presence of free electrons in CAPP. Prolonged plasma treatment of aqueous solutions of vitamin C, 1,4-hydroquinone, and 1,4-benzoquinone respectively, leads to their decomposition. Studies of the mechanisms in plasma-induced processes can help to prevent side effects in medicine, agriculture, and food disinfection.

## 1. Introduction

Cold atmospheric-pressure plasma (CAPP) is a quasi-neutral, partly ionized gas with space charge shielding operated at atmospheric pressures, consisting of atoms, electrons, ions and UV-Vis photons (Figure 1). CAPPs produce various radicals, ions, and atomic or molecular species when interacting with molecular gases such as air and with liquids such as water [1,2,3,4,5,6,7]. For example, CAPP can produce reactive oxygen and nitrogen species (RONS) [2,8] including radicals, ions, and relatively stable molecules such as HNO_3_, HNO_2_, NO_x_, H_2_O_2_, and O_3_ (Figure 1). These CAPP products lead to the activation of surface modifying processes [9,10,11,12,13]. There has been an increased number of applications of CAPP in agriculture and medicine within the last two decades [14,15], in which the application of cold plasma in agriculture for treatments of seeds, plants, flowers, and fruits is now called plasma agriculture [10,11,12,13]. Plasma medicine includes plasma treatments in dermatology [16,17], dentistry, skin cancer, disinfection [18,19], sterilization [20], drug delivery, and poration of bio-tissues [10,11].

Cold plasma can behave as a catalyst for the oxidation of nitrogen gas, occurring at the plasma/water and plasma/air interfaces [2,21,22,23]. Due to the reduced activation energy, atmospheric nitrogen can be converted to HNO_3_ and HNO_2_ which are useful for the production of nitrogen compounds and fertilizers [2]. Plasma treatment of water may provide a promising alternative to current methods of nitrogen fixation (for example, the thermal plasma Birkeland–Eyde process and the thermal-catalytic Haber–Bosch process), which have well-known environmental and ecological problems, such as high energy consumption at high temperatures and pressure with the emission of carbon dioxide.

RONS are known to participate in developmental processes by acting as signaling molecules for cell proliferation and differentiation, programmed cell death, seed germination, gravitropism, root hair growth, pollen tube development, and senescence. The use of cold plasma in food has proven to be an emerging technology. Cold plasma technology has been used in the food industry to reduce microbial count, degrade mycotoxin, inactivate enzymes, increase the concentration of bioactive compounds, and reduce allergens in food products.

Cold plasma has many biomedical applications [1,18,24,25]. These applications include acute and chronic wound healing, cleansing of dental cavities, surface activation of dental implants [26], improvement of conditions in infectious and inflammatory skin diseases, treatment of tumors and cancer, treatment of corneal infections, prevention of viral, fungal, and bacterial infections, and transdermal drug delivery [27] due to poration of the human skin. Poration is the formation of pores on a surface or a pattern of such pores. The reason may be that the magnitude of the electric fields generated by the plasma may exceed the threshold value for electroporation [10,11]. When applied to seed surfaces in agriculture, CAPP may also induce erosion, poration, and corrugation of the dormant seed to improve germination and water imbibition due to intracellular penetration of electric fields and RONS [10,11]. The use of cold atmospheric plasma in medicine and agriculture is very promising and effective but unfortunately, this method may have side effects. These include the effects of ultraviolet radiation, strong high-frequency electromagnetic fields, reactive oxygen and nitrogen radicals, ions, and molecules. Plasma can stimulate lipid peroxidation, oxidation of components of biological tissues, genotoxic effects, and formation of surface defects. The positive effects and side effects of cold plasma depend on the time of exposure to the biological tissue.

How does cold atmospheric pressure radio frequency plasma interact with water-soluble redox components of living cells? The most common examples in nature are water-soluble redox components such as NADH, NAD^+^, NADPH, NADP^+^, vitamin C (ascorbic acid), as well as hydrophilic and hydrophobic quinones in reduced or oxidized forms [28,29,30,31,32]. Can plasma negatively impact important biomolecules and thus lead to negative side effects of so far mostly positive treatment?

There are many publications about the side effects in biological tissues of various organisms in vivo when treated with cold plasma [9,12,13,33,34,35]. The purpose of this work is to elucidate in vitro the effect of plasma treatment when applied to individual water-soluble molecules that cause redox processes in living organisms.

## 2. Results

### 2.1. Oxygen Reduction, Nitrogen and Water Oxidation with Cold Atmosphere Pressure Radio-Frequency Plasma

It was demonstrated recently that cold plasma/water and plasma/air interfaces can have catalytic properties for charge-transfer reactions [2]. The majority of RONS produced by a plasma jet are unstable and have a very short lifetime. The most common and relatively stable products are HNO_3_, HNO_2_, H_2_O_2_, O_3_, and NO_x_ compounds. The majority of RONS, He^+^ cations, and radicals are strong oxidants. Figure 2 shows the thermodynamics (Gibbs energies) of all steps of nitrogen oxidation to HNO_3_. Electrochemical mechanisms and redox potentials of nitrogen and water oxidation at the cold atmospheric plasma/water interface were analyzed earlier [2]. The major product of this 5-electron redox reaction is HNO_3_ and small impurities of less stable compounds HNO_2_ and H_2_O_2_ are also produced. The process of aqueous phase acidification during nitrogen fixation starts at the interface between cold plasma and water [2].

### 2.2. Plasma Activated Water in the Presence of Water-Soluble Biologically Active Donors and Acceptors of Electrons NADH, NADPH, and NAD^+^

NAD^+^ is a coenzyme in single cells and multicellular organisms. NAD^+^ and NADH strongly absorb ultraviolet light because of adenine. The oxidized form of NAD^+^ absorbs light at a lower wavelength than the reduced form NADH (Figure 3 and Figure 4). The pyridine ring of NAD^+^ has an electronically more stable structure than the quinonoid form, hence NADH absorbs light at 340 nm whilst NAD^+^ does not (Figure 3 and Figure 4). Peak absorption of NAD^+^ is at a wavelength of 259 nm. NAD^+^ does not absorb light above 300 nm. NADH also absorbs at higher wavelengths with a second peak in UV absorption at 340 nm. Cold atmospheric pressure He-plasma jet (CAPPJ) does not reduce NAD^+^ to NADH during one hour of treatment (Figure 3A). H_2_O_2_, nitrate and nitrite test strips were used for detection of chemical reaction products, but not for their exact concentration. Chemical analysis of products shows that the presence of NAD+ (Figure 3) or NADH in the aqueous solution does not influence nitrogen fixation or the production of H_2_O_2_. Oxidation of NADH or NADPH leads to acidification of the aqueous phase.

NADH oxidation and NAD^+^ reduction are 2-electron reactions that can occur as two consecutive one-electron reactions or as a direct two-electron process:NADH ↔ NAD^+^ + H^+^ + 2e^−^(1)

CAPPJ oxidizes NADH to NAD^+^ (Figure 4A). The NADH oxidation rate was 47 μM/h. CAPPJ oxidizes NADPH to NADP^+^ (Figure 5A) with a speed of oxidation equal to 35 μM/h. The redox reaction in the NADPH/NADP^+^ couple is also a 2-electron process:NADPH ↔ NADP^+^ + H^+^ + 2e^−^(2)

### 2.3. Oxidation and Decomposition of Sodium Ascorbate, 1,4-Hydroquinone, and 1,4-Benzoquinone with Cold Atmospheric Pressure He-Plasma Jet

Ascorbic acid (AA) is a water-soluble antioxidant. Ascorbic acid in aqueous solutions is reversibly oxidized to dehydroascorbic acid (DHA) (Figure 6) and then irreversibly hydrolyzed to 2,3-diketo-L-gulonic acid (DKG) [36] which decomposes to L-xylose, L-xylonic, L-lyxonic, L-threonic, and oxalic acids (Ox) (Figure 6B). Cold plasma can strongly oxidize ascorbic acid in an aqueous solution (Figure 6), but in biological tissue or a multicomponent mixture, such as natural apple juice, the rate of the oxidation process can be reduced.

Ascorbic acid can transfer a single electron, because of the resonance-stabilized nature of its radical ion, called semidehydroascorbate (AR). Ascorbic acid can also transfer two electrons to acceptors synchronously without formation of the intermediate semidehydroascorbate radical ion [37,38,39]. A cold atmospheric pressure He-plasma jet oxidizes the ascorbate ion (1) to dehydroascorbate (2) as it is shown in Figure 6. Ascorbate has a maximum molar absorptivity at 265 nm.

Hydroquinone and semiquinone can participate in disproportionation, autoxidation, and cross-oxidation reactions. Hydroquinone participates in vivo in metabolic reactions of methylation, sulfation, or formation of glucuronides, turning them into biologically inactive compounds that are eliminated from the body.

Absorption spectra of 1,4-hydroquinone and 1,4-benzoquinone before and after treatment with cold He-plasma are shown in Figure 7. Hydroquinone has a maximum absorption of 288 nm (Figure 7A). Treatment of aqueous solutions with CAPPJ induces oxidation and decomposition of 1,4-hydroquinone without the formation of 1,4-bezoquinone which has a maximum absorption in water at 244 nm (Figure 7B).

Cold plasma does not reduce 1,4-benzoquinone but can oxidize it (Figure 7B). It is known in organic chemistry that both benzoquinone and hydroquinone can be oxidized by RONS to different compounds such as 2,3-epoxy-p-benzoquinone, benzoic acids, etc. When peroxide nucleophiles are attached to 1,4-benzoquinone and 1,4-hyroquinone, the formation of quinone epoxides occurs. This is typical for the H_2_O_2_-dependent oxidation of quinone’s double bonds, or in reactions of quinones with O_2_^•−^.

## 3. Discussion

The cold atmospheric pressure He-plasma consists of fast ionization waves propagating along the noble gas channel with speeds of a few hundred m/s. Electrons inside the fast ionization wave have energies of a few eV and are capable of producing ionization and non-equilibrium chemical reactions at room temperature [1]. The reactive species are generated by these electrons, He^+^ cations, and UV radiation at the plasma/air and plasma/water interfaces (Figure 1).

Cold atmospheric pressure He-plasma jet propagating into ambient air and forming RONS can induce oxidation of the most common water-soluble redox compounds in multicellular organisms, such as NADH, NADPH, and L(+)-ascorbic acid sodium salt (vitamin C) at interfaces between air, cold atmospheric pressure plasma, and water (Figure 4, Figure 5 and Figure 6). CAPPJ is unable to reduce NAD^+^ and 1,4-benzoquinone (Figure 3 and Figure 7) despite the presence of free electrons in CAPP. Prolonged plasma treatment of aqueous solutions of vitamin C (Figure 6), 1,4-hydroquinone, and 1,4-benzoquinone (Figure 7) leads to their decomposition. A CAPP can induce different heterogeneous and homogeneous reactions between oxygen, ozone, nitrogen, and water, which include production of HNO_x_ at the plasma/air and plasma/water interfaces as well as within the plasma-activated water [2]. Among these redox reactions, the most important are
N_2_ + O_2_ → NO_x_   at the plasma/air interface(3)
4NO + 2H_2_O + O_2_ → 4HNO_2_ → 4NO_2_^−^ + 4H^+^  at the plasma/water interface(4)
2HNO_2_ + O_2_ → 2HNO_3_ → 2NO_3_^−^ + 2H^+^     at the plasma/water interface(5)
2NO + O_2_ → 2NO_2_^−^   at the plasma/air interface(6)
4NO_2_ + 2H_2_O + O_2_ → 4HNO_3_ → 4NO_3_^−^ + 4H^+^  at the plasma/water interface(7)

Another possible process of the acidification of an aqueous phase could be the oxidation of water at the plasma/water interface:2H_2_O → H_2_O_2_ + 2H^+^ + 2e^−^(8)

Redox reactions (1, 2, 4, 5, 7, 8) produce acidification of an aqueous phase. Cold atmospheric pressure He-plasma jet produces acidification of an aqueous phase, oxidation of water-soluble compounds in the reduced form, such as NADH, NADPH, and L(+)-ascorbic acid. CAPPJ can induce decomposition of water-soluble redox compounds such as L(+)-ascorbic acid (Figure 7B), 1,4-hydroquinone, and 1,4-benzoquinone (Figure 8).

There are many beneficial effects of short low dose CAPPJ applications on bio-tissues (Table 1, left column), but it’s important to recognize possible side effects of high dose cold plasma applications (Table 1, right column).

Prolonged treatment of biological tissues can lead to side effects such as peroxidation, oxidation, acidification, decomposition, denaturation of bio-tissues, genotoxic problems, and DNA damage. Genotoxicity of CAPP refers to the capability of UV and RONS produced by cold plasma to damage the genetic information of cells.

## 4. Materials and Methods

### 4.1. Chemicals and Test Strips

β-Nicotinamide adenine dinucleotide reduced disodium salt hydrate (NADH), β-nicotinamide adenine dinucleotide sodium salt hydrate (NAD+), β-nicotinamide adenine dinucleotide 2′-phosphate reduced tetrasodium salt hydrate (NADPH), 1,4-hydroquinone, 1,4-benzoquinone, and L(+)-ascorbic acid sodium salt (vitamin C sodium salt) were purchased from Sigma-Aldrich (USA). H_2_O_2_, nitrate, and nitrite test strips were purchased from Bartovation LLC (New York, NY, USA). These strips were used for the detection of H_2_O_2_, NO_2_^−^, and NO_3_^−^ in water after the treatment with cold atmospheric pressure He-plasma jet. Time of the aqueous solution treatment was shown in Figure 3. Bottled ultra-high purity helium was purchased from Sexton Welding Supply (Huntsville, AL, USA).

### 4.2. Plasma Source

The plasma was powered with a high voltage pulsed DC system consisting of a Matsusada AU-10P60 10 kV DC power supply (Matsusada Precision Inc., Shiga, Japan), an IXYS PVX-4110 pulse generator (DEI, Fort Collins, CO, USA), and a DG-1022Z digital function generator (Rigol Technologies Inc., Beaverton, OR, USA). The system was operated with an 8 kV pulse amplitude, 6 kHz pulse frequency, 1 μs pulse width, and ~70 ns pulse rise and fall time. The experimental setup was described earlier [2,11,13]. Helium gas was flown at a rate of 2 L/min in the annulus between the two tubes.

Each sample of aqueous solutions was thoroughly stirred with a magnetic stirrer during treatment with cold atmospheric pressure He-plasma jet to ensure a homogeneous composition of chemicals in the 20 mL clear borosilicate glass vial (Figure 8). The diameter of the water surface was 15 mm.

### 4.3. Temperature Control

The digital laser temperature gun, Etekcity lasergrip 800 (Etekcity, Anaheim, CA, USA), was used for the measurement of temperature in the plasma jet, water, and air. The temperature of the aqueous phase during 60 min of plasma treatment was 20 °C.

### 4.4. Absorption Spectra

Absorption spectra were recorded using 1 cm quartz cuvettes with the Shimadzu UV-Vis spectrophotometer ISR-2600 Plus (Shimadzu, Japan).

### 4.5. Statistical Analysis

The software SigmaPlot 12 (Systat Software, Inc., Chicago, IL, USA) was used for statistical analysis of experimental data. All experimental results were reproduced at least 14 times.

## 5. Conclusions

Cold atmospheric pressure radio frequency plasma (CAPP) is widely used in medicine, agriculture, food processing for the treatment of various diseases, and disinfection of biological tissues and food due to its antibacterial, antiviral, and antifungal properties. The mechanisms of these multicomponent processes in biological tissues remain partially elucidated. CAPP in air can produce reactive oxygen and nitrogen species (RONS) including radicals, ions, and relatively stable molecules. Cold atmospheric pressure plasma can induce oxidation of the most common water-soluble redox compounds in multicellular organisms, such as NADH, NADPH, L(+)-ascorbic acid sodium salt (vitamin C), and quinones at interfaces between three fluid phases: air, cold atmospheric pressure plasma, and water. CAPPJ is unable to reduce NAD^+^ and 1,4-benzoquinone despite the presence of free electrons in CAPP. Plasma is used in biomedical research directly on multicomponent biological tissues [46,47]. To clarify the mechanisms, it is necessary to know both the mechanisms of interaction of plasma with individual components in vitro and the mechanisms of interaction with several components in a mixture. Plasma oxidation of ascorbic acid has been detected in other works, but it can much less degrade in mixture with other antioxidant components in biological objects [46]. Studies of the mechanisms of plasma-induced processes in biological tissues and surfaces can help to neutralize or prevent side effects in animal and human medicine, cosmetology, agriculture, and the food industry.

## Figures and Tables

**Figure 1 molecules-27-07051-f001:**
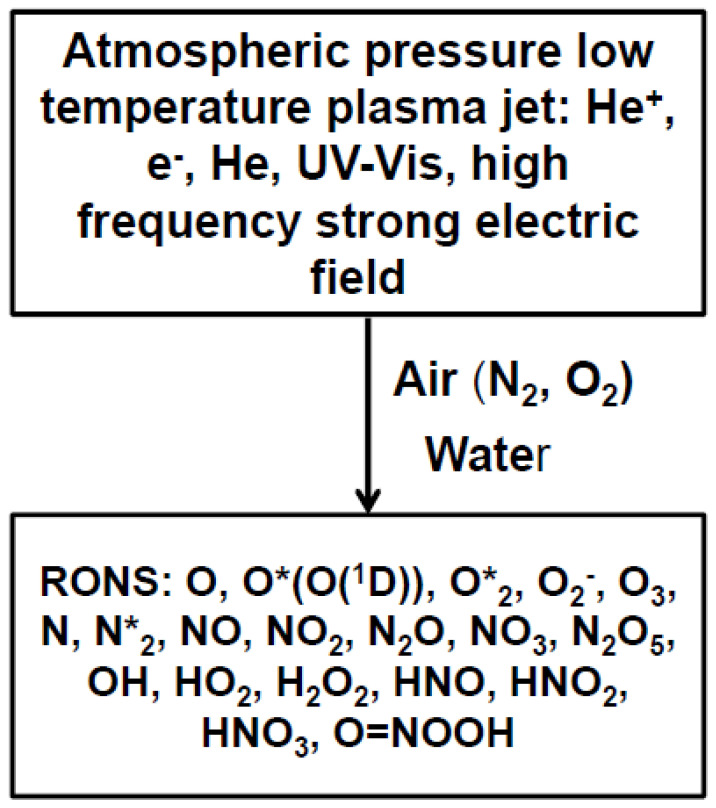
Diagram of RONS production with cold atmospheric pressure radio-frequency He-plasma jet at interfaces between air, plasma, and water.

**Figure 2 molecules-27-07051-f002:**
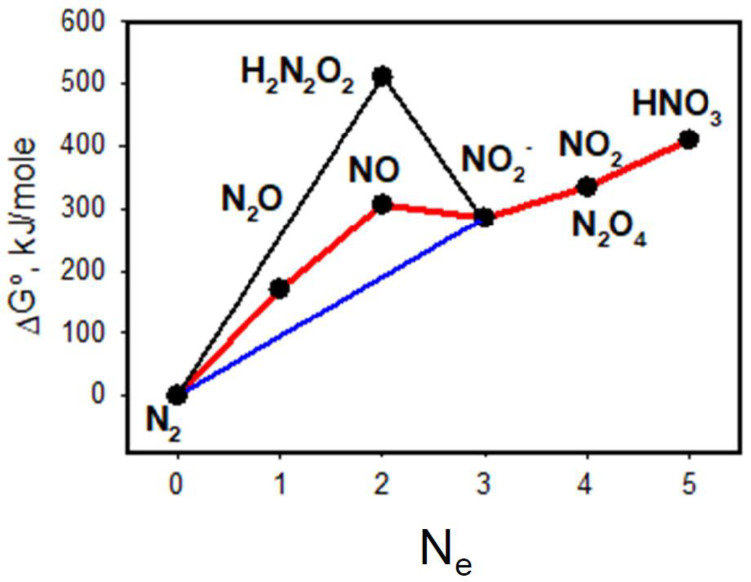
Gibbs energy of N_2_ multielectron oxidation at the CAPPJ/water interface; N_e_ is the number of electrons. The red line shows the most probable pathway of nitrogen oxidation to HNO_3_.

**Figure 3 molecules-27-07051-f003:**
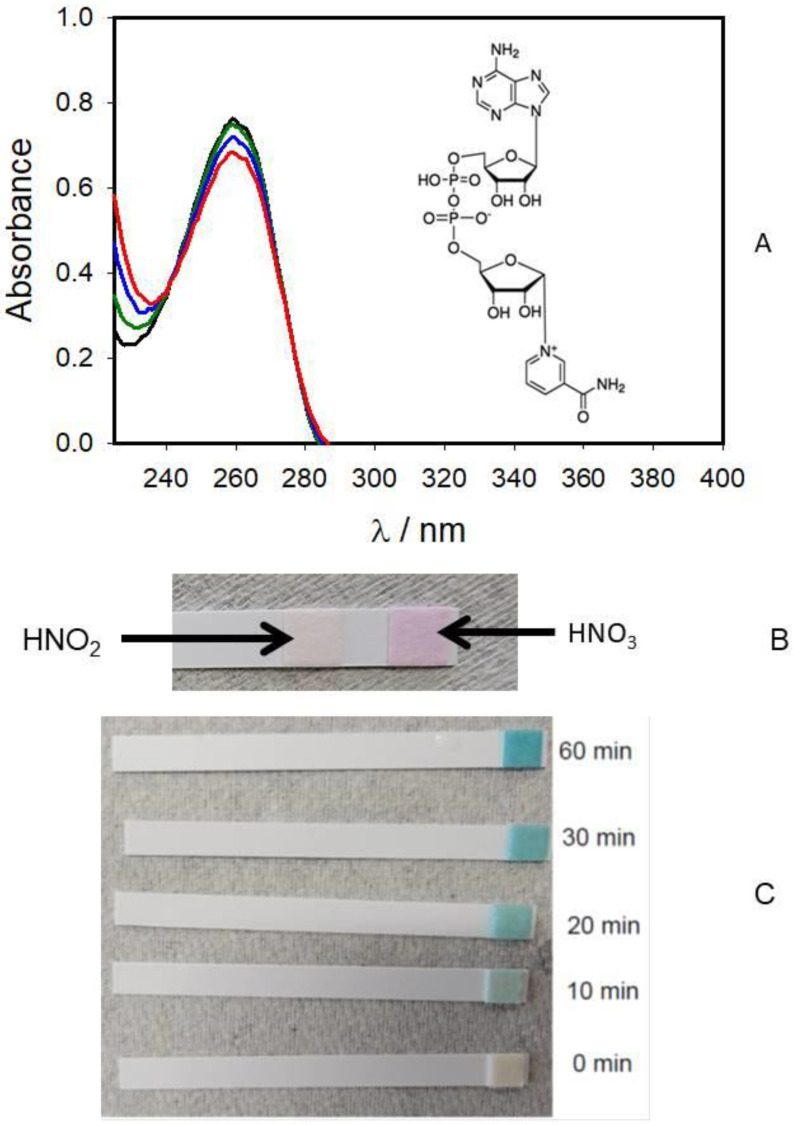
(**A**) Absorption spectra of 50 μM NAD^+^ aqueous solution before (black) and after the treatment with cold atmospheric pressure He-plasma jet for 10 min (green), 30 min (blue), and 60 min (red). (**B**) Testing with nitrate (≈0.2 mM) and nitrite (≈0.01 mM) white test strips the production of HNO_3_ and HNO_2_ after 60 min treatment of 50 μM NAD^+^ aqueous solution with cold atmospheric pressure He-plasma jet. (**C**) Detection with test strips H_2_O_2_ production during the treatment of 50 μM NAD^+^ aqueous solution with cold atmospheric pressure He-plasma jet. The test strips acquire specific color only in the presence of specific substrates, such as HNO_3_ and HNO_2_ (**B**) or H_2_O_2_ (**C**). The temperature was 20 °C.

**Figure 4 molecules-27-07051-f004:**
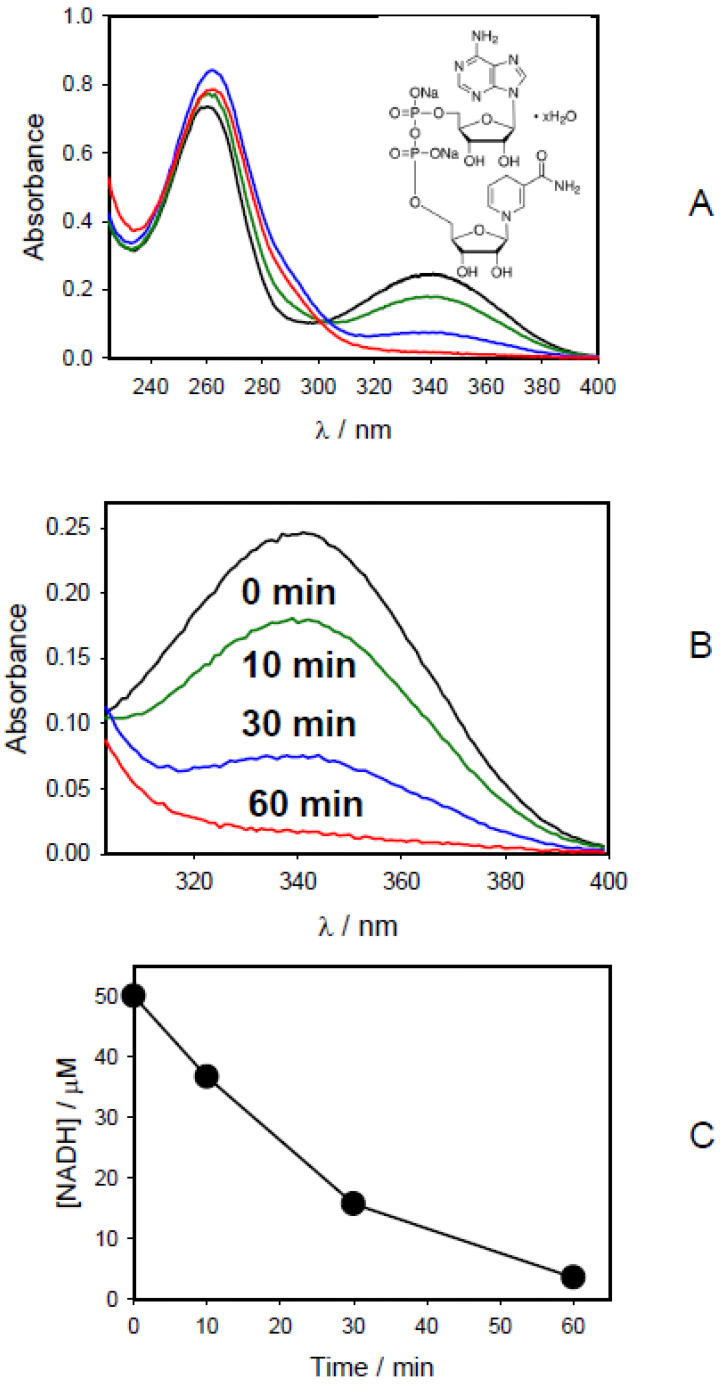
(**A**,**B**) Absorption spectra of 50 μM NADH aqueous solution before (1) and after treatment with cold atmospheric pressure He-plasma jet during 10 min (2), 30 min (3), and 60 min (4). (**C**) Dependence of NADH concentration during the time of CAPPJ calculated from the absorption spectra shown in Figure 4B and a calibration curve.

**Figure 5 molecules-27-07051-f005:**
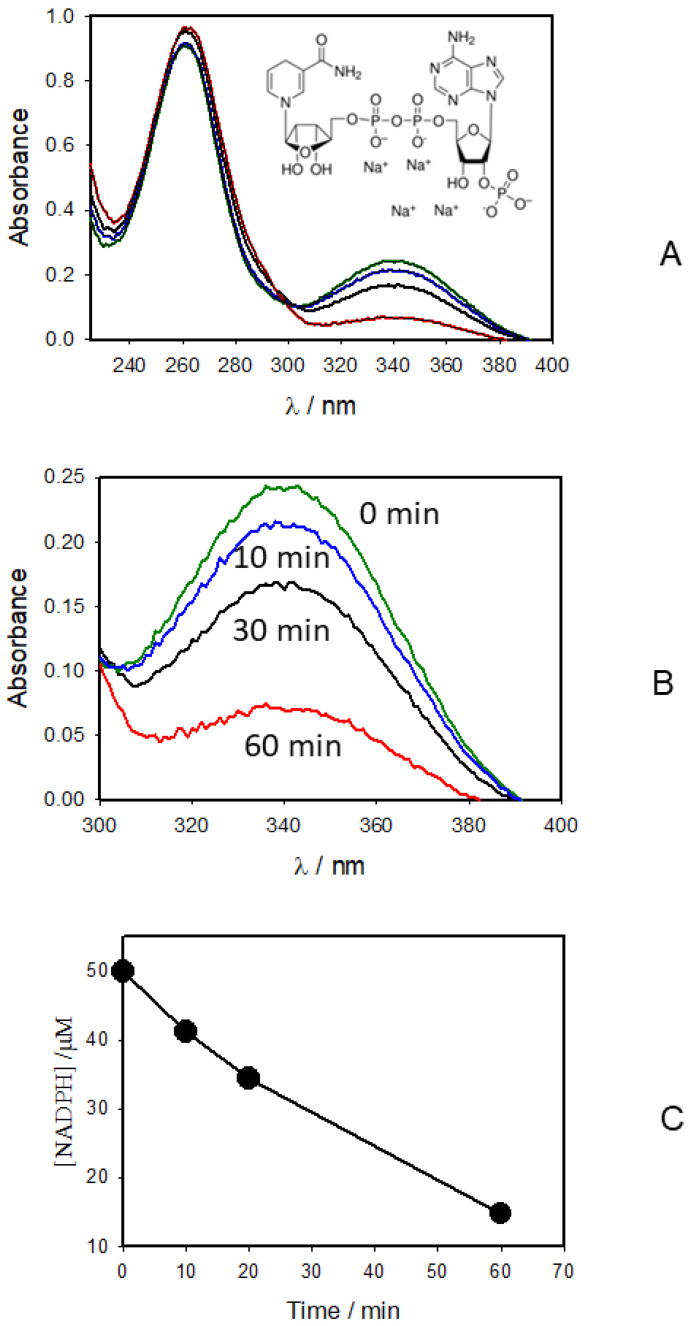
(**A**,**B**) Absorption spectra of 50 μM NADPH aqueous solution before (green) and after treatment with cold atmospheric pressure He-plasma jet for 10 min (blue), 30 min (black), and 60 min (red). (**C**) Dependence of NADPH concentration on the CAPPJ irradiation time calculated from the absorption spectra shown in Figure 5B and a calibration curve. The temperature was 20 °C.

**Figure 6 molecules-27-07051-f006:**
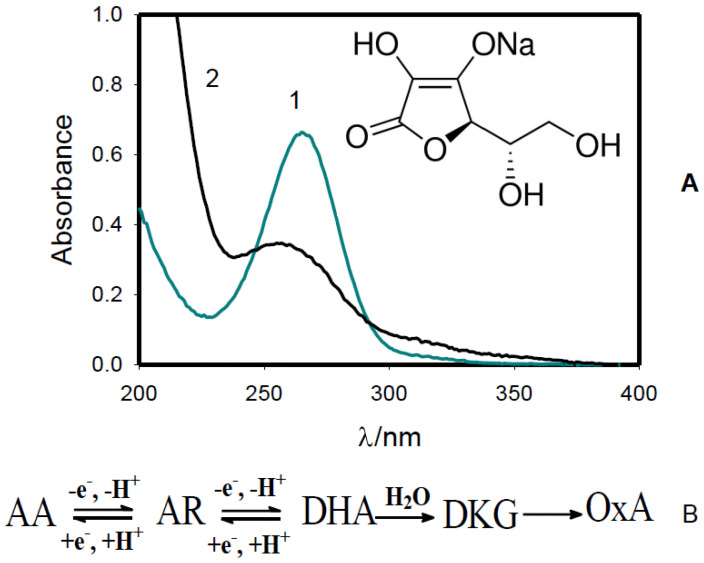
(**A**) Absorption spectra of 0.25 mM sodium ascorbate aqueous solution before (1) and after treatment with cold atmospheric pressure He-plasma jet for 30 min (2). (**B**) Oxidation, hydrolysis, and decomposition processes of sodium ascorbate.

**Figure 7 molecules-27-07051-f007:**
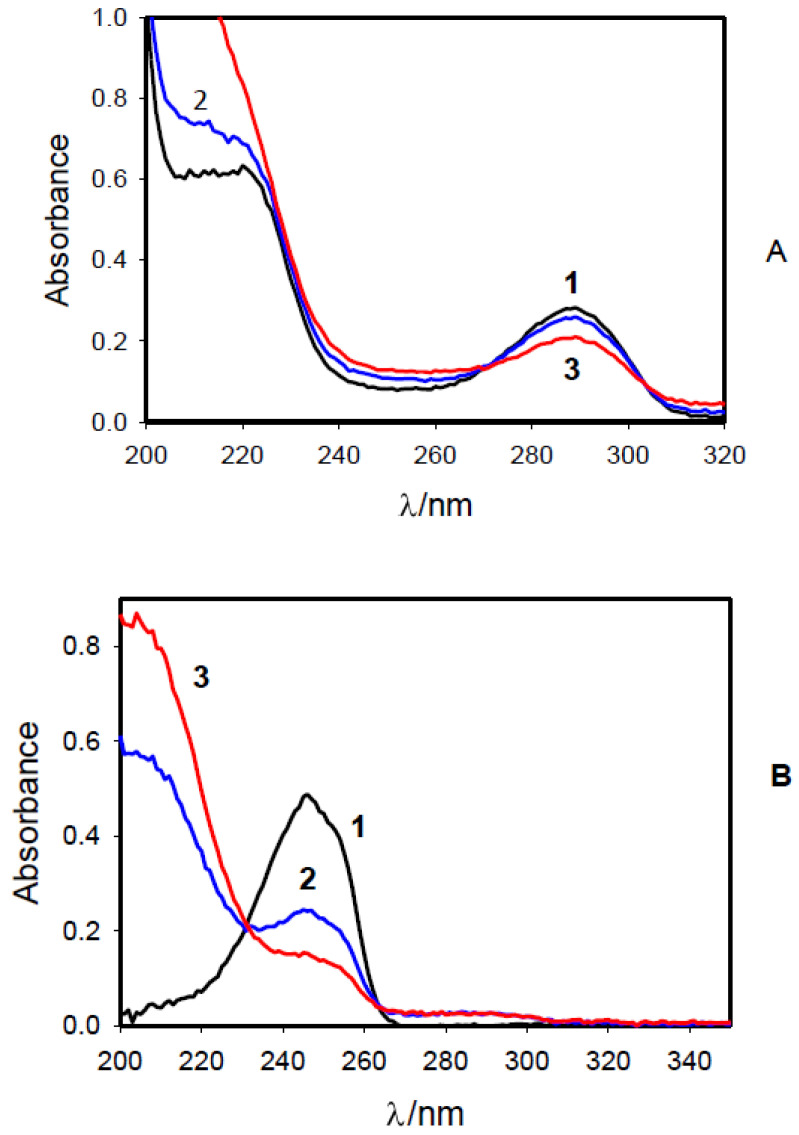
Absorption spectra of 50 μM 1,4-hydroquinone (**A**) and 1,4-benzoquinone (**B**) aqueous solution before (1) and after treatment with cold atmospheric pressure He-plasma jet for 30 min (2), and 60 min (3). The temperature was 20 °C.

**Figure 8 molecules-27-07051-f008:**
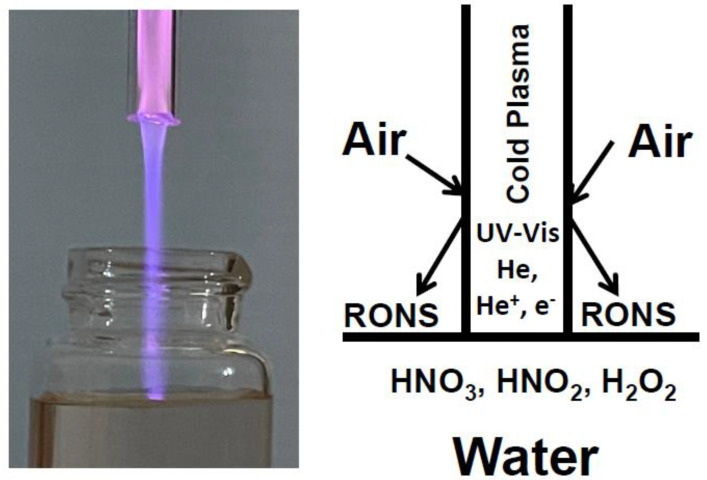
Plasma/water, plasma/air, and air/water interfaces. 20 mL clear borosilicate glass vials with magnetic stirrers were used in all experiments with CAPPJ treatments. The diameter of the water surface was 15 mm. The temperature was 20 °C.

**Table 1 molecules-27-07051-t001:** Cold atmospheric pressure plasma has beneficial and side effects on biological tissue.

Beneficial Effects of Low Dose Cold Plasma Applications	Possible Side Effects of High Dose Cold Plasma Applications
Disinfection and sterilization of bio-tissue [1]; Acceleration of plant seeds imbibition, germination, and growth [10,11]; Increasing crop yields [10,11]; Erosion, poration, and corrugation of bio-tissue surfaces and membranes [10,11,12,13]; Activation of ion channels [9,12]; Activation of specific signaling pathways [12,13]; Catalysis of redox reactions [2]; Wound healing [24]; Cleaning dental cavities, activation of dental implants [26,27]; Treatments in dermatology [16,17,18,25]; Treatments in cosmetology [25]; Cancer and tumor treatment [18]; Protection from bacteria, viruses, and fungi [19,20]; Food treatment [35]; Corneal infections treatment [20]; Activation of defense hormones and gene expression [14].	Oxidation of water- soluble and hydrophobic antioxidants (NADH, NADPH, vitamin C, quinones, etc.); Decomposition of water-soluble redox compounds such as L(+)-ascorbic acid, 1,4-hydroquinone, and 1,4-benzoquinone; RONS and UV damaging effects [9,40]; Genotoxic effects [33,34]; Genetic modification of organisms [33,40]; Generation of strong electrical signals in bio-tissue [9,27]; Change of electrical fields in bio-tissue [9]; Membrane breakdown [18]; Interaction with enzymatic systems [34]; Acidification of bio-tissue [2]; Peroxidation of lipids and bio-tissue [33]; Deactivation, oxidation or denaturation of enzymes [34]; Negative immune responses; Blood coagulation [41,42]; Cell damage and death [43]; Necrosis [41]; Structural changes, oxidation and modification of amino acids, proteins and DNA [44,45].

## Data Availability

The data presented in this study are available on request from the corresponding author.

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
