# Peer review of "Redox Reactions of Biologically Active Molecules upon Cold Atmospheric Pressure Plasma Treatment of Aqueous Solutions"

_molecules, 2022, doi:10.3390/molecules27207051_

Round 1

Reviewer 1 Report

Alexander G. Volkov and colleagues submitted a manuscript entitled „Redox reactions of biologically active molecules upon cold atmospheric pressure plasma treatment of aqueous solutions“ for publication in molecules.

The authors used NADH, NADPH, ascorbic acid as water soluble biologically active molecules ubiquitously present in living organisms as well as hydro-and benzoquinone. The authors used absorption spectra of single substances and detected oxidative changes after plasma treatment. Further decomposition of sodium ascorbic acid, hydroquinone and benzoquinone by plasma jet application in liquids was detected. These oxidative changes were discussed in regard to described side effects of plasma applications such as DNA damage and lipid peroxidation.

The introduction is very broad and introduces plasma applications in many life science fields ranging from agriculture, seed germination, pore formation, dentistry to medicine and cancer treatment. It is not clear how the named topics are specifically connected to the presented results. Only the last section proposes some questions, such as „how plasma interacts with these redox compounds in multicellular organisms?“ to formulate the objective of the study. The following experimental approach is rather simple by comparison. The redox compounds in question were exposed to plasma over a period of time (up to 60 min) and their absorption spectrum was recorded. The measurement was performed in water. The relevance to a biological system might be questionable, since no determination took place in a multicellular organism or in the presence of cultured cells (as a biological test system). Neither were different compounds mixed. Oxidative effects are measurabel. Unfortunately, the biological effects or consequences are only inferred not shown on a test system and appaer therefore in parts far fetched. The authors claim that the oxidation of NADH and NADPH by CAP causes undesired side effects of CAP applications. In nature there are always more than one system for balancing oxidative impacts. Moreover, in a multicomponent system several reaction partners are present and will react with plasma-generated ROS. It is well documented that NADH is required to balance oxidative stress and protect cellular structures. The authors span a huge loop from simple oxidation of redox compounds in solution to a range of side effects of high dose plasma applications (Table1) without really showing one experimental proof of it. The link is missing between the detected oxidation of water soluble antioxidants, the proposed chemical reactions and the named biological consequences. Perhaps the authors could offer some more specific impact chains where these compounds play a role in a biological system.

In my opinion, the questions raised in the introduction “How CAP interacts with water soluble redox components in multicellular organisms?“ and „Can plasma negatively impact important biomolecules and thus lead to negative side effects of so far mostly positive treatment?“ were not addressed in this work. Or the aim needs to be rephrased.

Are absorption spectra useful and meaningful in multicomponent systems? If there are several reaction partners available in a solution the oxidation capacity of CAP diverts of all partners and may be very different compared to solutions with one component. How does the findings apply to a multicomponent mixtures?

The authors adress the different phases of plasma liquid interaction where reactions take place. The role for the present reactions is not clearly represented. The compounds in question are solved in water and reactions occur in the liquid. Since the air remained stable in this experimental approach it is not obvious what influence the phases have. The presence of oxygen (and nitrogen) only seems to be mandatory.

The expression „interfaces between three fluid phases: air, cold atmospheric pressure plasma, and water“ implies fluids as phases but plasma is a gaseous phase. I suggest to use a different terminology to describe the different phases.

Line 109 …NAD+ is a coenzyme in all multicellular living cells. … It can either be multicellular or single cells . Please rephrase this sentence. 

Fig 4 B. The figure would be easier to understand if the treatment times were added to the graph instead of just numbers that need to be explained separetely in the legend text. Moreover, Fig 5 shows corresponding results for NADPH in aqueous solution. but here the labelling is missing (Fig 5B).

Line 127-128… How was the reaction rate/oxidation rate calculated? Standard curve of NAD+?  It is not described. Please add a brief description to materials and methods for better understanding. Further there is a descrepancy between the legend text and the graph in Fig 5C. The time points do not match (20 or 30 min). Please check the graph.

Author Response

Answers to the Reviewers: We wish to thank the editors and reviewers for reading and commenting on our paper. We have taken into consideration all their suggestions to revise and to improve our manuscript.

Reviewer #1

Statement: The authors used NADH, NADPH, ascorbic acid as water soluble biologically active molecules ubiquitously present in living organisms as well as hydro-and benzoquinone. The authors used absorption spectra of single substances and detected oxidative changes after plasma treatment. Further decomposition of sodium ascorbic acid, hydroquinone and benzoquinone by plasma jet application in liquids was detected. These oxidative changes were discussed in regard to described side effects of plasma applications such as DNA damage and lipid peroxidation.

Answer: Thank you for your kind comments.

Statement: The introduction is very broad and introduces plasma applications in many life science fields ranging from agriculture, seed germination, pore formation, dentistry to medicine and cancer treatment. It is not clear how the named topics are specifically connected to the presented results. Only the last section proposes some questions, such as „how plasma interacts with these redox compounds in multicellular organisms?“ to formulate the objective of the study. The following experimental approach is rather simple by comparison. The redox compounds in question were exposed to plasma over a period of time (up to 60 min) and their absorption spectrum was recorded. The measurement was performed in water. The relevance to a biological system might be questionable, since no determination took place in a multicellular organism or in the presence of cultured cells (as a biological test system). Neither were different compounds mixed. Oxidative effects are measurabel. Unfortunately, the biological effects or consequences are only inferred not shown on a test system and appaer therefore in parts far fetched. The authors claim that the oxidation of NADH and NADPH by CAP causes undesired side effects of CAP applications. In nature there are always more than one system for balancing oxidative impacts. Moreover, in a multicomponent system several reaction partners are present and will react with plasma-generated ROS. It is well documented that NADH is required to balance oxidative stress and protect cellular structures. The authors span a huge loop from simple oxidation of redox compounds in solution to a range of side effects of high dose plasma applications (Table1) without really showing one experimental proof of it. The link is missing between the detected oxidation of water soluble antioxidants, the proposed chemical reactions and the named biological consequences. Perhaps the authors could offer some more specific impact chains where these compounds play a role in a biological system.

Answer: In the introduction section everything is correct. We have mentioned importance of plasma in different multicomponent processes in chemistry, biology, medicine, agriculture, and nitrogen fixation. It is very important to study reaction between individual biologically active molecules and plasma for understanding mechanisms of plasma induced redox processes in bio-tissue and evaluation of possible side effects. Our work on processes in multicomponent solutions induced by cold plasma is in progress and some of results were already published in references [2, 9-13]. Table 1 in the Discussion section summarizes our results and our knowledge from literature with exact references.

Statement: In my opinion, the questions raised in the introduction “How CAP interacts with water soluble redox components in multicellular organisms?“ and „Can plasma negatively impact important biomolecules and thus lead to negative side effects of so far mostly positive treatment?“ were not addressed in this work. Or the aim needs to be rephrased.

Answer: The aim in the introduction is: “There are many publications about the side effects in biological tissues of various organisms in vivo when treated with cold plasma [9,12,13,33-35]. The purpose of this work is to elucidate in vitro the effect of plasma treatment when applied to individual water-soluble molecules that cause redox processes in living organisms.”

Statement: Are absorption spectra useful and meaningful in multicomponent systems? If there are several reaction partners available in a solution the oxidation capacity of CAP diverts of all partners and may be very different compared to solutions with one component. How does the findings apply to a multicomponent mixtures?

Answer: The Reviewer is right. Probably, the treatment of mixture of different components with cold plasma can show different kinetics of oxidation of some components depending on redox potentials. We have comments about it in our Conclusions section: “Plasma is used in biomedical research directly on multicomponent biological tissues [46,47]. To clarify the mechanisms, it is necessary to know both the mechanisms of interaction of plasma with individual components in vitro and the mechanisms of interaction with several components in a mixture. Plasma oxidation of ascorbic acid has been detected in other works, but it can much less degrade in mixture with other antioxidant components in biological objects [46].”

Statement: The authors address the different phases of plasma liquid interaction where reactions take place. The role for the present reactions is not clearly represented. The compounds in question are solved in water and reactions occur in the liquid. Since the air remained stable in this experimental approach it is not obvious what influence the phases have. The presence of oxygen (and nitrogen) only seems to be mandatory.

Answer: We found earlier [2] using indicators that reactions between plasma, air and aqueous solutions take place at the aqueous interface, but not in the bulk of liquid phase. The presence of RONS, water, oxygen, nitrogen and plasma are mandatory.

Statement: The expression „interfaces between three fluid phases: air, cold atmospheric pressure plasma, and water“ implies fluids as phases but plasma is a gaseous phase. I suggest to use a different terminology to describe the different phases.

Answer: According to physics and physical chemistry, air, plasma and water are fluid phases and 3 different states of a matter [1,2,18]. Follow to your suggestion, we modified our sentence in the abstract: “Here we found that CAPP can induce in vitro oxidation of the most common water-soluble redox compounds in living cells such as NADH, NADPH, and vitamin C at interfaces between: air, CAPP, and water.”

Statement: Line 109 …NAD+ is a coenzyme in all multicellular living cells. … It can either be multicellular or single cells . Please rephrase this sentence.

Answer: This sentence was rephrased: “NAD+ is a coenzyme in single and multicellular living cells.”

Statement: Fig 4 B. The figure would be easier to understand if the treatment times were added to the graph instead of just numbers that need to be explained separetely in the legend text. Moreover, Fig 5 shows corresponding results for NADPH in aqueous solution. but here the labelling is missing (Fig 5B).

Answer: Figure 5 has labeling in the Legend to Figure: ”(A, B) Absorption spectra of 50 mM NADPH aqueous solution before (green) and after treatment with cold atmospheric pressure He-plasma jet for 10 min (blue), 30 min (black), and 60 min (red).”

The treatment times were added inside Figs. 4B and 5B according to the Reviewer advice.

Statement: Line 127-128… How was the reaction rate/oxidation rate calculated? Standard curve of NAD+?  It is not described. Please add a brief description to materials and methods for better understanding. Further there is a descrepancy between the legend text and the graph in Fig 5C. The time points do not match (20 or 30 min). Please check the graph.

Answer: The graph is correct. We added additional information to Legends to Figures:

“(C) Dependence of NADH concentration during the time of CAPPJ calculated from the absorption spectra shown in Fig. 4B and a calibration curve”.

“(C) Dependence of NADPH concentration on the CAPPJ irradiation time calculated from the absorption spectra shown in Fig. 5B and a calibration curve”.

Reviewer 2 Report

It is an up-to-date study on the effect of cold plasma on some compounds found in living organisms. The results are interesting and the strengths are represented by the clear and concise presentation of both the results and the work method. Also the figures are edifying. This field of research is topical and the results of this study are of transdisciplinary interest. I think that the method should specify whether the tests were inserted into the glass vessel during the plasma treatment or after the treatment and how long the water treatment was done.

In this study, the authors tested in vitro the effect of cold plasma treatment on some ubiquitous molecules in living organisms (NADH, NADPH, NAD+, 1,4-hydroquinone, 1,4-benzoquinone, ascorbic acid). The experimental protocol consisted in the cold plasma treatment of the salt solutions of the above molecules and the recording of the absorption spectra. The results showed that cold plasma treatment causes the oxidation of NADH, NADPH, and vitamin C, while prolonged exposure causes the decomposition of vitamin C, 1,4-hydroquinone, and 1,4-benzoquinone. The study is interesting and useful for the field of biochemistry studies regarding the effect of cold plasma on biological molecules ubiquitous in living organisms. However, it must be taken into account that the study was carried out in vitro, therefore my recommendation is to emphasize this aspect both in the abstract and in the results and discussions. I state this, since the abstract does not state that the study is in vitro, and the discussions based on the results obtained in this study are speculative concerning the leaving organisms. In vitro research is extremely important, but it is known that the effects on living organisms may or may not be similar due to their complexity. My recommendations are to state in the abstract that the study is in vitro. The figures should have the same quality and clarity. Attention to formulas L 116 "H2O2". The working method must be improved by specifying the concentration of the salt solution used in the study and the method of calculating the concentration of NADH in fig 4C and NDPH fig 5C, was a calibration curve used?''

Author Response

Statement: “It is an up-to-date study on the effect of cold plasma on some compounds found in living organisms. The results are interesting and the strengths are represented by the clear and concise presentation of both the results and the work method. Also the figures are edifying. This field of research is topical and the results of this study are of transdisciplinary interest. I think that the method should specify whether the tests were inserted into the glass vessel during the plasma treatment or after the treatment and how long the water treatment was done.”

Answer: Thank you for your kind comments.

Statement: “I think that the method should specify whether the tests were inserted into the glass vessel during the plasma treatment or after the treatment and how long the water treatment was done.”

Answer: We have included the following information in the revised manuscript:  “These strips were used for the detection of H2O2, NO2-, and NO3- in water after the treatment with cold atmospheric pressure He-plasma jet. Time of the aqueous solution treatment was shown in Fig. 3.”

Statement: “In this study, the authors tested in vitro the effect of cold plasma treatment on some ubiquitous molecules in living organisms (NADH, NADPH, NAD+, 1,4-hydroquinone, 1,4-benzoquinone, ascorbic acid). The experimental protocol consisted in the cold plasma treatment of the salt solutions of the above molecules and the recording of the absorption spectra. The results showed that cold plasma treatment causes the oxidation of NADH, NADPH, and vitamin C, while prolonged exposure causes the decomposition of vitamin C, 1,4-hydroquinone, and 1,4-benzoquinone. The study is interesting and useful for the field of biochemistry studies regarding the effect of cold plasma on biological molecules ubiquitous in living organisms.”

Answer: Thank you for your kind comments.

Statement: “However, it must be taken into account that the study was carried out in vitro, therefore my recommendation is to emphasize this aspect both in the abstract and in the results and discussions. I state this, since the abstract does not state that the study is in vitro, and the discussions based on the results obtained in this study are speculative concerning the leaving organisms. In vitro research is extremely important, but it is known that the effects on living organisms may or may not be similar due to their complexity. My recommendations are to state in the abstract that the study is in vitro. The figures should have the same quality and clarity.”

Answer: We have stated your recommendations in the abstract, introduction, and in discussions:

Abstract, L. 20: “Here we found that CAPP can induce in vitro oxidation of the most common water-soluble redox compounds…”

Introduction, L. 87: “The purpose of this work is to elucidate in vitro the effect of plasma treatment when applied to individual water-soluble molecules that cause redox processes in living organisms.”

Conclusions, L. 290-294: “Plasma is used in biomedical research directly on multicomponent biological tissues [46,47]. To clarify the mechanisms, it is necessary to know both the mechanisms of interaction of plasma with individual components in vitro and the mechanisms of interaction with several components in a mixture. Plasma oxidation of ascorbic acid has been detected in other works, but it can much less degrade in mixture with other antioxidant components in biological objects [46]. Studies of the mechanisms of plasma-induced processes in biological tissues and surfaces can help to neutralize or prevent side effects in animal and human medicine, cosmetology, agriculture, and the food industry.”

Statement: “The figures should have the same quality and clarity. Attention to formulas L 116 "H2O2".”

Answer: Thank you. We corrected our typo and changed “H2O2” to “H2O2”.

Statement: “The working method must be improved by specifying the concentration of the salt solution used in the study and the method of calculating the concentration of NADH in fig 4C and NDPH fig 5C, was a calibration curve used?''

Answer: Concentrations of all redox compounds are shown in Legends to Figures. We  have included the following information in the revised manuscript:

“(C) Dependence of NADH concentration during the time of CAPPJ calculated from the absorption spectra shown in Fig. 4B and a calibration curve”.

“(C) Dependence of NADPH concentration on the CAPPJ irradiation time calculated from the absorption spectra shown in Fig. 5B and a calibration curve”.

Round 2

Reviewer 1 Report

The authors provided a point by point reply to all statements raised in the first review report. They also amended the manuscript in parts and where it was adviced.

Please check and correct the chemical structure in Fig.4A. There seems to be an excessive phosphate group.

Author Response

Statement: “The authors provided a point by point reply to all statements raised in the first review report. They also amended the manuscript in parts and where it was adviced.”

Answer: Thank you for your kind comments.

Statement: Please check and correct the chemical structure in Fig.4A. There seems to be an excessive phosphate group.

Answer: We have corrected the chemical structure in Fig.4A in accordance with the Reviewer's comment.